# Health professionals' readiness and its associated factors to implement Telemedicine system at private hospitals in Amhara region, Ethiopia 2021

Sisay Maru Wubante[ID]*, Araya Mesfin Nigatu, Adamu Takele Jemere

Department of Health Informatics, Institute of Public Health, College of Medicine and Health Sciences, University of Gondar, Gondar, Ethiopia

* Sisay419@gmail.com

## Abstract

### Introduction

In resource-limited settings incorporating the Telemedicine system into the healthcare system enhances exchanging valid health information for practicing evidence-based medicine for the diagnosis, treatment, and prevention of diseases. Despite its great importance, the adoption of telemedicine in low-income country settings, like Ethiopia, was lagging and increasingly failed. Assessing the readiness of health professionals before the actual adoption of telemedicine is considered the prominent solution to tackle the problem. However, little is known about Health professionals' telemedicine readiness in this study setting.

### Objective

Accordingly, this study aimed to assess health professionals' readiness and its associated factors to implement a Telemedicine system at private hospitals in North West, Ethiopia.

### Materials and methods

An institution-based cross-sectional study was conducted from March 3 to April 7, 2021. A total of 423 health professionals working in private hospitals were selected using a simple random sampling technique. Multi-variable logistic regression was fitted to identify determinant factors of health professional readiness after the other covariates were controlled.

### Result

In this study the overall readiness of telemedicine adoption was 65.4% (n = 268) [95% CI:60.1–69.8]. Knowledge (AOR = 2.5;95% CI: [1.4, 4.6]), Attitude (AOR = 3.2;95% CI: [1.6, 6.2]), computer literacy (AOR = 2.2; 95% CI: [1.3, 3.9]), computer training (AOR = 2.1;95% CI: [1.1, 4.1]), Computer skill (AOR = 1.9;95% CI: [1.1, 3.4]), computer access at office (AOR = 2.1;95% CI: [1.1, 3.7]), Internet access at office (AOR = 2.8; 95% CI: [1.6, 5.1]), Own personal computer (AOR = 3.0; 95% CI: [1.5, 5.9]) and work experience (AOR = 3.1;

**Data Availability Statement:** All relevant data are within the manuscript and its Supporting Information files.

**Funding:** The authors received no specific funding for this work.

**Competing interests:** The authors have declared that no competing interests exist.

**Abbreviations:** ICT, Information Communication Technology; eHealth, Electronic health; SPSS, statistical package for social sciences; AOR, Adjusted odd ratio.

95% CI: [1.4, 6.7]) were significantly associated with the overall health professionals readiness for the adoption of telemedicine using a cut point of p-value lessthan 0.05.

## Conclusion and recommendation

Around two-thirds of the respondents had a good level of overall readiness for the adoption of telemedicine. The finding implied that less effort is required to improve readiness before the implementation of telemedicine. This findings implied that respondents who had good knowledge and a favorable attitude toward telemedicine were more ready for such technology. Capacity building is needed Enhance computer literacy, and computer skills building their confidence to rise ready for such technology. Building their capacity through training, building good internet connection, and availability of computers, where the necessary measures to improve Telemedicine readiness in this setting. Additionally, further studies are recommended to encompass all types of telemedicine readiness such as organizational readiness, technology readiness, societal readiness, and so on. Additionally, exploring the healthcare provider opinion with qualitative study and extending the proposed study to other implementation settings are recommended to be addressed in future works. The study has a positive impact on the successful implementation and use of telemedicine throughout hospitals at countries level by providing pertinent information about health professionals' preparedness status. Therefore, implementing telemedicine will have a significant contribution to the health system performance improvement in terms of providing quality care, accessibility to health facilities, reduction of costs, and creating a platform for communication between health professionals across different health institutions for providing quality patient care.

## Introduction

Information and Communication Technology (ICT) is being used to improve health care quality and allow patients to access primary healthcare through digital technologies modality, in different countries in the world [1]. Digital technologies including telemedicine have plenty of benefits to healthcare delivery with the help of good telecommunication infrastructures, information technology infrastructures, medical terminologies, and trained health professionals it helps to provide health services and manage patients at distance [2]. In the era of covid-19 pandemics, telemedicine have a great role in controlling the pandemics by using different communication modalities including, online consultation, telemonitoring/screening and chatbox and sensors, and geographic position system to avoid potentially dangerous location [3].

Introducing a telemedicine system to the healthcare system provides a platform for health professionals to exchange valid and quality health information about diagnosis treatment and management of patients and to apply evidence-based practice and its saves both health professionals and patients time and costs [4, 5]. Despite the potential of telemedicine to improve the quality of patient care significant proportion fails to support patient management in low-income countries [6].

Different countries in the world introduced telemedicine to their health systems to improve the health quality of their populations and modernize their health system, however many of them were not successfully implemented [7].

More than 75% of telemedicine system projects failed without significant contributions to the health system globally [8]. However developed countries are better to implement

telemedicine as compared with low-income nations, 76% of united states of America health institutions full function the system [9]. 75% of Norway's health institutions successfully implement telemedicine system, But in low-income countries only 10%of their health institutions provide health service through telemedicine [8, 10].

To successfully implement a telemedicine system readiness assessment is a significant factor for the adoption and utilization of telemedicine system health institutions and is fundamental [11, 12].

Readiness assessment finds outs existing conditions of health institutions and health professionals' preparedness for a new system.

Preparedness evaluation refers to the preparedness of healthcare systems and health professionals to adopt changes that have been brought about by the implementation of computerized systems [13]. The readiness assessment of healthcare providers has been identified as a critical success factor in the implementation of electronic systems [14].

Research done in Lebanon on health professionals' readiness toward electronic health reported that the majority of study participants were ready to implementation of electronic health [15]. A study conducted in Uganda on health professionals' readiness toward telemedicine found that 64.7% of health professionals were ready to implement telemedicine [16]. Another study conducted in Mauritius on health professionals' readiness to implement telemedicine showed the readiness of health professionals for Telemedicine system implementation was 64.5% [17].

Multiple reasons are given for low adoption of telemedicine systems such as resistance to change, lack of pre-implementation preparation, lack of organizational readiness, unavailability of information technology, and lack of technical skill of personnel [18–21].

The successful introduction of a telemedicine system into healthcare organizations requires the study of different technical, organizational infrastructures, and human factors [15]. The study revealed that health professionals' readiness is fundamental for launching a telemedicine system in healthcare [22]. Examining readiness is an inclusive measurement to know about possible reasons for telemedicine adoption failure and provide pertinent information on existing conditions and the preparedness of health care organizations to change.

Like other countries, Ethiopia's ministry of health recognized the importance of information technology to the health system is indicated and tried to implement telemedicine pilot projects at some selected hospitals, but failed without net contribution to the health system. To fully implement electronic health at the national level electronic health strategy was developed including five domains including telemedicine, mobile health, electronic learning, electronic medical records, and health information system. The government of Ethiopia has made different efforts to implement an electronic health information system including a district health information system version2 for reporting system and electronic medical records and telemedicine for a few hospitals as a pilot and plans to scale up in the future.

To researcher knowledge limited research is done in Ethiopia, assessing health professionals' readiness towards telemedicine has paramount significance to the health system.

These findings help input for policymakers, program managers, and local health administrators to launch a full system and sustain it for the future.

Therefore, this study aimed assessed health professionals' readiness toward telemedicine and its associated factors working at private hospitals Amhara region, North West Ethiopia.

## Materials and methods

The Institutional Review Board of the University of Gondor, College of Medicine and Health Sciences, Institute of Public Health, issued an ethical clearance with the reference number

(RfNo/IPH/1493/2013). Verbal consent was obtained, and study subjects were told that they were able to complete the questionnaire suggested as their contract to engage in the study. During previous research, we found that many of the study subjects were unwilling to sign a written authorization form. This factor, along with the overwhelming number of respondents, led us to resort to verbal consent. The ethics procedure clarified this, and the ethics board precluded the signed consent requirement. As a result, rather than offering written approval, we chose to read and describe the consent form to respondents during data collection.

## Study design and settings

An institution-based cross-sectional quantitative approach was conducted from March 3 to April 7, 2021. The study was conducted at private hospitals in the Amhara region of North West Ethiopia. The capital city of the Amhara region is bahirdar.

## Source and study population

Private hospitals found in the Amhara region of Ethiopia are general hospitals.gamby general hospital has 205 health professionals and about 165 beds. Adinas general hospital has about 73 health professionals and 45 beds. Dream care general hospital has 67 health professionals and 38 beds. Afilias general hospital has about 38 health professionals and 23 beds. Ibex general hospital has about 34 health professionals and 22 beds.ayu general hospital has 41 health professionals and 25 beds. Bati general hospital has 58 health professionals and 26 beds. Selam general hospital has 81 health professionals and 35 beds.yifat general hospital has 41 health professionals and 21 beds. Ethio general hospital has 61 health professionals and 29 beds.

In this study source and study population are the same because ten private hospitals were included in the study. All health professionals who were working at ten private Hospitals in the Amhara Region of Ethiopia were considered as the source and study of the population.

## Inclusion criteria

Private hospitals found in the Amhara region of Ethiopia are general hospitals.gamby general hospital has 30 specialists. Adinas the general hospital has 23 specialists. Dearm care general hospital has 20 specilialist. Afilias general hospital has 12 specialists. Ibex general hospital has 15 specialists.ayu general hospital has 11 specialist.bati general hospital has 13 specialists.selam general hospital has 19 specialists. Yifat general hospital has 10 specialists. Ethio general hospital has 9 specialists. All Health professionals who were working at ten private Hospitals and available at the time of data collection were included in the study.

## Exclusion criteria

Health professionals who were seriously ill and those who have less than six months of working experience in clinical practices were excluded from the study. Because they are new to the work environment and may not have knowhow the application of telemedicine technology for healthcare delivery.

## Study participants, sample size, and sampling procedure

Health professionals working at private hospitals found in the Amhara region of northwest Ethiopia were eligible for this study. The sample size was calculated by using the single population proportion formula by considering 50%with a 95% level of confidence, a 5% margin of error a 10% non-response rate. Finally, a minimum sample size of 423 was obtained.

The following formula was used to calculate the total sample size of the study.

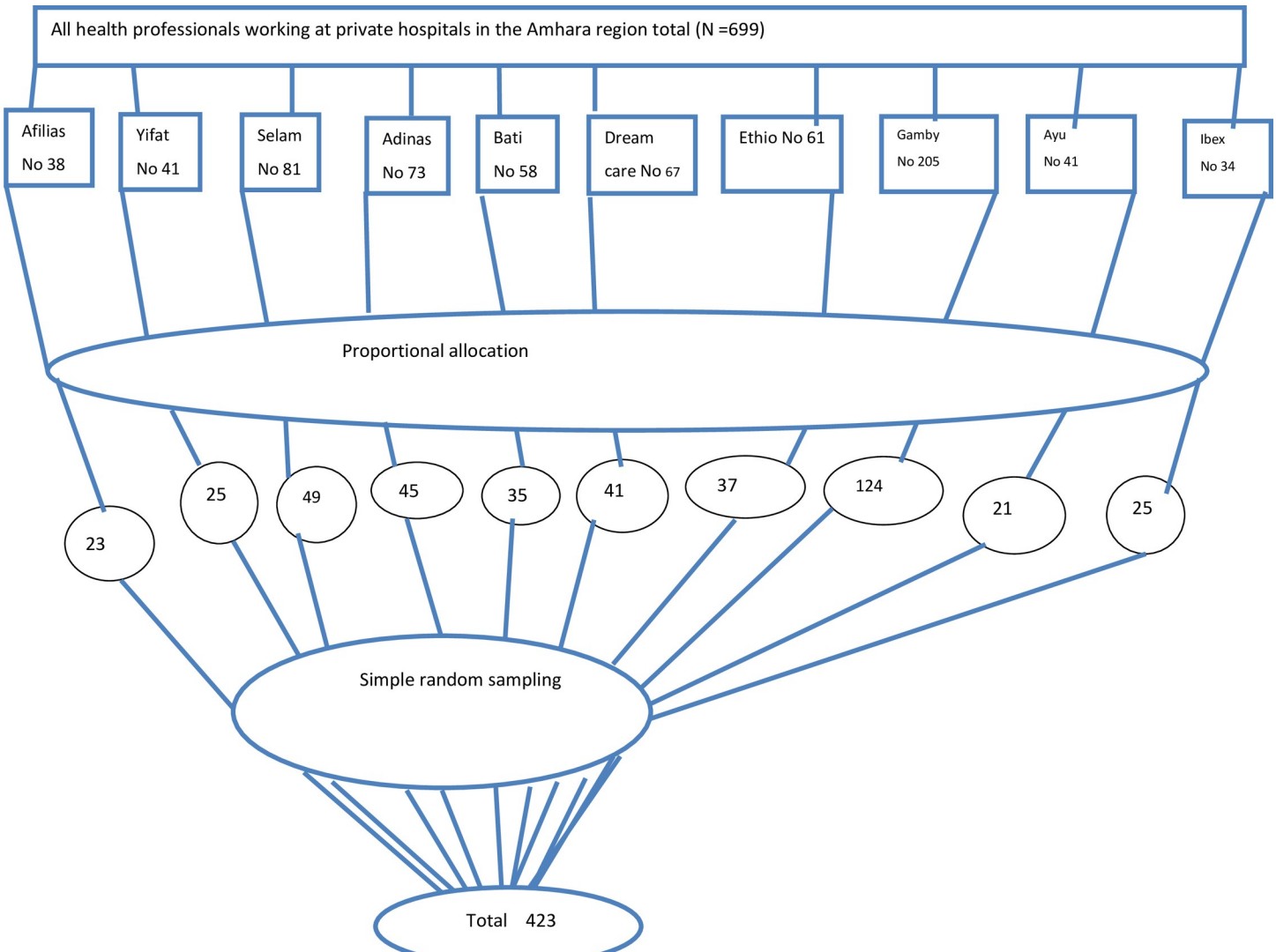

**Fig 1. Sampling procedure for selecting study participants at private hospitals in Amhara region, Ethiopia 2021.**

$$\text{Sample size (n)} = \frac{(Z^{\alpha/2})^{2 \text{ x p}(1-p)}}{d^2}, \quad (n) = \frac{(n) = (1.96)2 \text{ x } 0.5(1-0.5) = 384.2 \text{ x } 10\% = 423}{(0.05)2}$$

proportional allocation was done for each hospital and study participants were recruited by a simple random sampling technique (Fig 1).

## Data collection tool and procedures

Data were collected using a pretested and structured self-administered questionnaire adapted from [13]. The first part of the questionnaire contents, demographic information, individual, technical, and organizational access to basic technologies variables were included in the questionnaire and it includes yes and Likert scale questions. The second part includes questions on core and engagement readiness. Core readiness was examined by health professionals' dissatisfaction with the current paper-based system(four items), while engagement readiness was

assessed by health professionals based on the potential benefits and willingness to use the telemedicine system (eleven items). These two readiness domains were measured s in a five-point Likert scale with one denoting strongly disagree to five indicating strongly agree. The English version of the questionnaires is translated to the local language Amharic The local language is also translated to English. The reliability was also checked using Cronbach alpha's coefficient (overall Cronbach alpha for health professionals readiness = 0.81).

Health informatics professionals were recruited as supervisors and eight health information technician professionals participated in data collection. To control the quality of data, two days of training were given to data collectors and supervisors on the objective of the study, data collection procedures, data collection tools, respondents' approach, data confidentiality, and respondents right before the data collection date. Before the actual data collection, pretesting of the questionnaire was conducted for about 10%of the study participants.

## Measurements

In this study, core readiness and engagement readiness were assessed using a set of composite scores that included four core questions and eleven engagement questions. The realization of needs and expressed dissatisfaction with the current way of working, such as inefficient documentation, patient privacy, and data confidentiality, is defined as core readiness [1]. Engagement Readiness is defined as an active willingness to implement telemedicine and volunteer to receive telemedicine training [23].

In this study, we used the mean and median scores to dichotomize our variables such as Telemedicine readiness, knowledge, attitude, computer literacy, and computer skill. If the variable was normally distributed, we computed the mean score. On the other hand, we used the median score if the variable was not normally distributed.

**Health professionals' readiness.**   In this study health professionals' readiness was measured by a set of 15 questions and participants who scored median and above were considered ready, and participants who scored below the median were considered as not ready [24].

**Knowledge.**   Respondents' level of knowledge of telemedicine was assessed by questions to be answered in either "Yes" or "No." A score of "1" will be given for "Yes" and "0" for "No." One can score a minimum of 0 and a maximum of 18 in this section. In this study, the average score of 9 (50%) from the 18 questions was used as a cutoff point to determine the level of knowledge of telemedicine. The mean knowledge score of less than 9 (50%) was labeled as poor knowledge of telemedicine and the more than average score of 9 (50%) was labeled as good knowledge of telemedicine [25].

**Attitude.**   The respondents' attitude was assessed by questions answered by rated on a 5-point Likert scale that ranged from "1 = strongly disagree" to "5 = strongly agree". In this study, a mean score of less than 2.5 (50%) is labeled as poor attitude, and 2.6 and more (51%)– 3.0 (60%) is labeled as favorable attitude [26].

## Data processing and analysis

Before entry to Epi info and exported to SPSS The data were manually checked for completeness and consistency. The data were entered into the Epi info version7.2 and exported to SPSS version 20 for analysis. Descriptive statistics were used to describe the characteristics of study participants in terms of socio-demographic and other variables. Categorical variables were presented in form of a frequency table to describe study subjects. Study subjects' preparedness was measured by calculating the overall scores readiness level. The median scores for each readiness component were evaluated to find the overall readiness levels, and respondents who scored below or above the median overall readiness score were regarded as not ready (No) or

ready (Yes) for Telemedicine implementation, respectively. to determine the association between independent variables such as gender educational status, work experiences, knowledge, attitude, computer literacy, etc, the dependent variable readiness was analyzed by multivariable logistic regression. Multicollinearity was also checked to manage the confounding effect of each variable. In multivariable logistic regression model analysis, model fitness was checked by the Hosmer-Lemeshow test (0.052) Variable's significant association was determined based on the adjusted odds ratio (AOR), with 95% Cl and variables with (p<0.05) were considered as determinant factors for health professional readiness towards the implementation of Telemedicine.

## Result

### Socio-demographic characteristics of study participants

In this study, a total of 423 health professionals from ten private hospitals were approached, out of them, 410 responded with a response rate of 96.9%. More than half 226 (55.1%) of the study participants were males. The majority (44.6%) of the respondents were within the age group 25–29 years (Table 1).

### Technical related factors

Of the total study participants, More than half 240 (58.5%) of health professionals were computer-literate. About 245 (59.8%) of study participants had computer skills (Table 2).

### Organizational related factors

About 245 (59.8%) of study participants have had computer access at their offices. Regarding, internet access at the office nearly two-thirds of 251 (61.2%) of the health professionals had internet access at their office (Table 3).

**Table 1. Socio-demographic characteristics of health professionals working in all private hospitals Amhara region 2021 (N = 410).**

| Variables | Categories | Frequency (N) | Percentage (%) |
|---|---|---|---|
| Sex | Male | 226 | 55.1 |
| | Female | 184 | 44.9 |
| Age | 20–24 | 29 | 7.1 |
| | 25–29 | 183 | 44.6 |
| | 30–34 | 128 | 31.2 |
| | > = 35 | 70 | 17.1 |
| Professions | Medical Doctor | 90 | 22.0 |
| | Nurse | 165 | 40.2 |
| | midwifery | 40 | 9.8 |
| | pharmacy | 46 | 11.2 |
| | Medical laboratory | 57 | 13.9 |
| | Other | 12 | 2.9 |
| Work experiences | <2years | 66 | 16.1 |
| | 2–3 years | 58 | 14.1 |
| | 4–5 years | 67 | 16.3 |
| | >5 years | 219 | 53.4 |
| Educational status | Diploma | 82 | 20.0 |
| | Degree | 202 | 49.3 |
| | Masters and above | 126 | 30.7 |

**Table 2. Technical factors towards telemedicine readiness among health professionals at private hospitals in the Amhara region 2021.**

| Variables | Category | Frequency (N) | Percent (%) |
|---|---|---|---|
| Computer literate | Yes | 240 | 58.5 |
| | No | 170 | 41.5 |
| Computer skill | Yes | 245 | 59.8 |
| | No | 165 | 40.2 |

### Behavioral related factors

In this study, more than two-thirds of 281 (68.5%) of the study participants had good knowledge. Similarly, half of 217 (52.9%) of the study participants had a favorable attitude toward Telemedicine (Fig 2).

### Health professional readiness to ward telemedicine system

Of a total of study participants, 244 (59.5%) and 222 (54.1%)of study participants have had core and engagement readiness respectively. The result of this study indicated that two-thirds of 268 (65.4%) with 95% CI: (60.1–69.8) study participants were ready for telemedicine adoption (Fig 3).

### Factors associated with telemedicine system readiness

As shown in (Table 4) variables in multivariable logistic regression like Knowledge, Attitude, own personal computer, internet access at the office, Computer skill, computer literacy, computer training, Work experiences, educational status, gender, and computer access at the office were positively associated with health professionals' readiness towards telemedicine.

The odds of readiness are 2.3 times high among health professionals who had good knowledge as compared with their counterparts. Respondents who had a favorable attitude toward Telemedicine were 3.2 times more likely ready for the Telemedicine system than those who have an unfavorable attitude. Having a personal computer was positively associated with health professionals' readiness for telemedicine adoption.

Computer skill was found significantly associated with respondents' readiness for telemedicine implementation. Computer literacy was statistically significant with health professionals' readiness for telemedicine adoption. Computer training was found positively associated with respondents' readiness for telemedicine adoption.

**Table 3. Organizational factors for health professionals' readiness towards telemedicine in private hospitals Amhara region 2021 (N = 410).**

| Variables | Categories | Frequency (N) | Percent (%) |
|---|---|---|---|
| Computer access at the office | Yes | 245 | 59.8 |
| | No | 165 | 40.2 |
| Internet access at the office | Yes | 251 | 61.2 |
| | No | 159 | 38.8 |
| Available IT support | Yes | 252 | 61.5 |
| | No | 158 | 38.5 |
| Computer Training | Yes | 168 | 41.0 |
| | No | 242 | 59.0 |
| Backup power generator | Yes | 334 | 81.5 |
| | No | 76 | 18.5 |

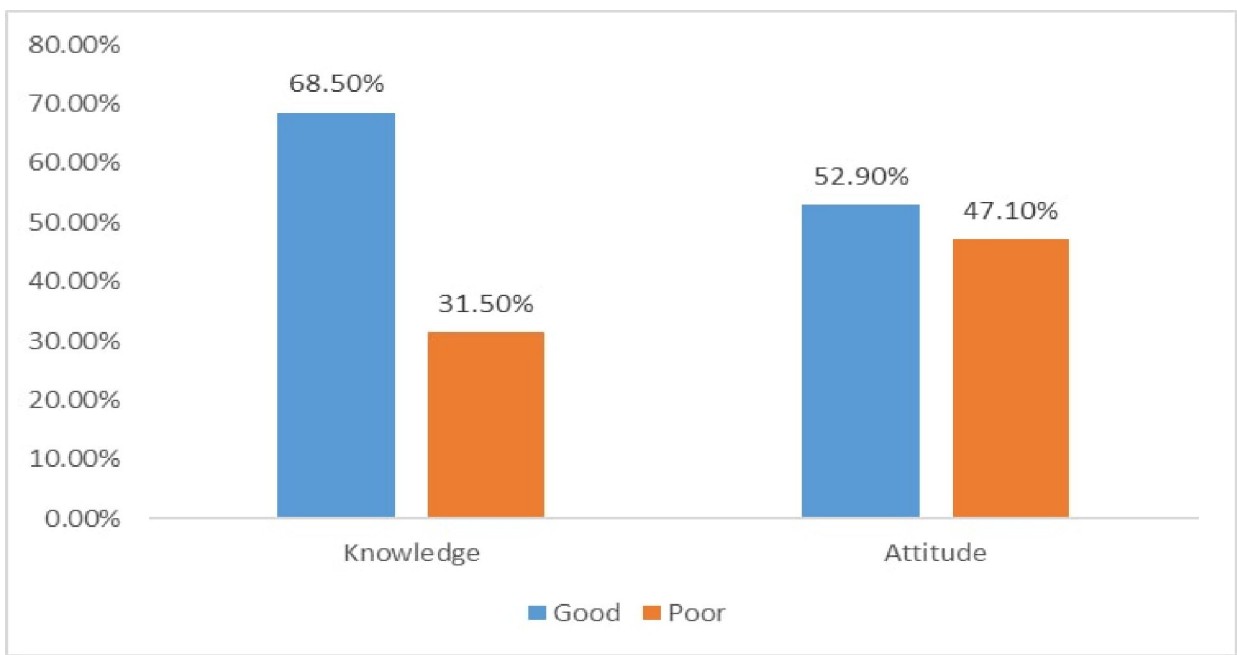

**Fig 2. Knowledge and attitude towards telemedicine among health professionals working at private hospitals in Amhara region, Ethiopia, 2021.**

## Discussion

The study assessed health professionals' readiness in private hospitals in Ethiopia for the planned national adoption of telemedicine. health professionals were targeted due to key determinants adoption and successful implementation of telemedicine depends on health professionals' readiness [27]. The readiness of healthcare professionals is a helpful component for the implementation successfully Telemedicine [28, 29]. To the best of our knowledge, no

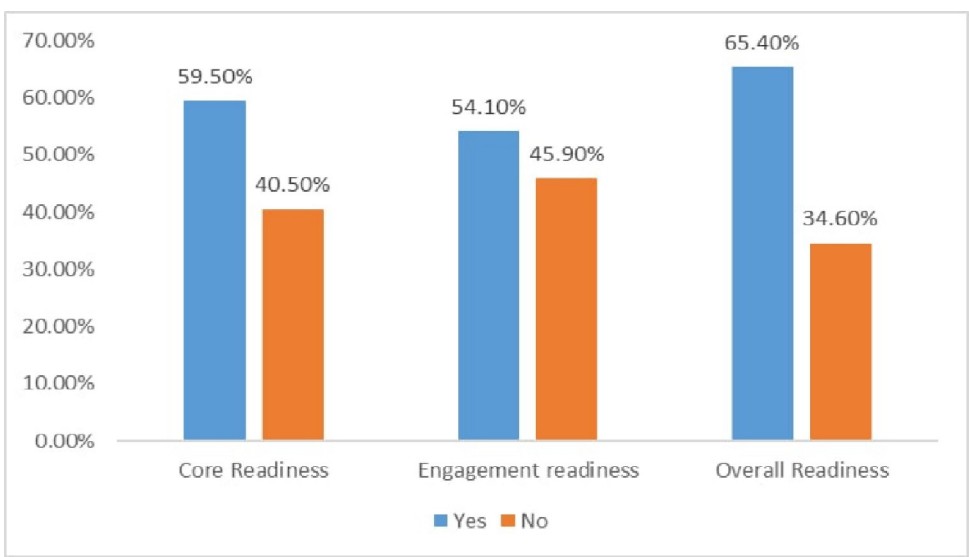

**Fig 3. Core and engagement and overall readiness of health professionals' towards telemedicine at private hospitals in Amhara region Ethiopia, 2021.**

**Table 4. Bi-variate and multivariate analysis on factors associated with readiness of health professionals for telemedicine system in private hospitals, Amhara region, 2021.**

| Variables | Readiness | | Crude OR (95% CI) | Adjusted OR (95% CI) | p-value |
|---|---|---|---|---|---|
| | Ready | Not ready | | | |
| Knowledge | | | | | |
| Good | 221 (78.6%) | 60 (21.4%) | 6.43 (4.1, 10.2) | 2.5 (1.4, 4.6)* | 0.002 |
| Poor | 47 (36.4%) | 82 (63.6%) | 1.0 | | |
| Attitude | | | | | |
| Favorable | 186 (85.7%) | 31 (14.3%) | 8.1 (5.1, 13.1) | 3.2 (1.6, 6.2)* | 0.001 |
| Unfavorable | 82 (42.5%) | 111 (57.5%) | 1.0 | | |
| Personal computer | | | | | |
| Yes | 223 (71.5%) | 89 (28.5%) | 2.96 (1.85, 4.71) | 3.0 (1.5, 5.9)* | 0.002 |
| No | 45 (45.9%) | 53 (54.1%) | 1.0 | | |
| Computer skill | | | | | |
| Adequate | 188 (76.7%) | 57 (23.3%) | 3.5 (2.3, 5.4) | 1.9 (1.1, 3.4)* | 0.025 |
| Not adequate | 80 (48.5%) | 85 (51.5%) | 1.0 | | |
| Computer literacy | | | | | |
| Adequate | 186 (77.5%) | 54 (22.5%) | 3.7 (2.4, 5.7) | 2.2 (1.3, 3.9)* | 0.007 |
| Not adequate | 82 (48.2) | 88 (51.8%) | 1.00 | | |
| Computer Training | | | | | |
| Yes | 140 (83.3%) | 28 (16.7%) | 4.5 (2.8, 7.2) | 2.1 (1.1, 4.1)* | 0.022 |
| No | 128 (52.9%) | 114 (47.1%) | 1.0 | | |
| Work experience | | | | | |
| <2 years | 35 (53%) | 31 (47%) | 1.0 | | |
| 2–3 years | 32 (55.2%) | 26 (44.8%) | 1.1 (0.5, 2.2) | 0.9 (0.4, 2.4) | |
| 4–5 years | 32 (47.8%) | 35 (52.2%) | 0.8 (0.4, 1.6) | 1.1 (0.4, 2.7) | |
| >5 years | 169 (77.2%) | 50 (22.8%) | 3.0 (1.9, 5.3) | 3.1 (1.4, 6.7)* | 0.004 |
| Internet access | | | | | |
| Yes | 200 (79.7%) | 51 (20.3%) | 5.3 (3.4, 8.2) | 2.8 (1.6, 5.1)* | 0.001 |
| No | 68 (42.8%) | 91 (57.2%) | 1.0 | | |
| Computer access | | | | | |
| Yes | 186 (75.9%) | 59 (24.1%) | 3.2 (2.1, 4.9) | 2.1 (1.1, 3.7)* | 0.017 |
| No | 82 (49.7%) | 83 (50.3%) | 1.0 | | |
| Sex | | | | | |
| Male | 171 (75.7%) | 55 (24.3%) | 2.8 (1.8, 4.2) | 1.5 (0.74, 2.9) | |
| Female | 97 (52.7%) | 87 (47.3%) | 1.0 | | |
| Educational status | | | | | |
| Diploma | 39 (47.6%) | 43 (52.4%) | 1.0 | | |
| Degree | 124 (61.4%) | 78 (38.6%) | 1.7 (1.0, 2.9) | 1.3 (0.58, 2.8) | |
| Master and above | 105 (83.3%) | 21 (16.7%) | 5.5 (2.9, 10.4) | 1.6 (0.7, 3.9) | |

studies have been undertaken in Ethiopia on health professionals' readiness for telemedicine system adoption.

In this study, the readiness of health professionals for the Telemedicine system in ten private hospitals that were frontline to implement the Telemedicine system was assessed. Our assessment indicated that the overall readiness of health professionals for the telemedicine system was 65.4%(with 59.5% of core readiness and 54.1% engagement readiness).

Nearly 59.5% were deemed ready for core readiness, while 54.1% were deemed ready for engagement readiness. This implies that health professionals may have expressed discomfort

with paper record systems and recognized the necessity of Telemedicine (core readiness), as they were seen as highly actively engaged with Telemedicine and willing to use it. assessing the core and engagement readiness of health professionals will help to know the advantage and disadvantages of telemedicine risk and the applicability of the system in healthcare institutions.

Health professionals' readiness in this result was consistent with readiness assessments done in other resource constraint settings in Uganda [16]. This is also consistent with a study conducted in Mauritius [17]. However slightly higher than another study done in Uganda [1]. This substantial difference could be due to infrastructure differences as well as differences in measurement methods used in the studies. Furthermore, higher education institutions in Ethiopia have currently integrated a generic health information system course for all health science students into their curriculum, which can provide information about the telemedicine system.

In this study health professionals who had good knowledge of Telemedicine, were about 2.5 times more likely to be ready for a Telemedicine system as compared to their counterparts. This finding is consistent with the previous study conducted in Mauritius [17]. Moreover, consistent with a study conducted in Iran [30]. This might be due to, health professionals with prior knowledge that may have helped their motivations to use the Telemedicine system. the speculation is per evidence suggested by another study [31], which confirmed that having a concept about Telemedicine improves readiness for the telemedicine system.

Health professionals who had a favorable attitude toward telemedicine were 3.2 more likely ready for the telemedicine system than their counterparts. This result is consistent with the study done in Saudi Arabia [32]. The reason is due to having a favorable view of telemedicine helps health professionals ready to adopt the telemedicine system.

This study showed that participants who owned personal computer were found to have telemedicine readiness.this is in line with a study done in Ethiopia [15, 33].

Computer skill was found to be significantly associated with telemedicine readiness. Participants who had good computer skill were 1.9 times more likely ready for telemedicine than their counterparts. This is consistent with studies done [1, 17, 33].

This might be because being familiar with computer technologies and having technical skills helps health professionals to be active adoption of telemedicine, not such difficult to adapt to new technology. Readiness to accept information systems is affected by the level of computer skill.

Computer literacy was found positively associated with telemedicine readiness. Respondents who had good computer literacy were 2.2 times more likely ready for telemedicine than their equivalents. This result is comparable with other studies done [34, 35]. The possible explanation might be knowing how to use computer technologies in day-to-day activities increase to use of advanced technologies.

Computer-related training was found significantly associated with telemedicine readiness. Health professionals who had taken computer-related training were about 2.1 times more likely ready for telemedicine systems than as compared with those health professionals who had not taken computer training. This is in line with a study done in Australia [35].

A possible reason for this could be computer training is more likely to increase participant familiarity with using technologies. Additionally, the explanation might be training and education usually changes people's views, and upgrade knowledge levels, and perceptions. Knowing the updated technology passionate for upcoming in their institution.

In this study internet access at the office was positively associated with telemedicine readiness. Health professionals who had internet access at their office were about 2.8 times more ready for the telemedicine system as compared with those health professionals who did not have internet access at the office. Consistent with a study done in [36]. This might be because

the internet influences access how new advanced technologies applications in the health system. Internet exposure can impact humankind's daily life.

Health professionals who had computer access at their office were about 2.1 times more ready for the telemedicine system as compared with those health professionals who did not have computer access at the office. This finding is supported by a previous study done in Lebanon [15]. This could be attributed to the availability of computer-enabled individuals to practice digital tools. Furthermore, computer access allows for a daily practice of Telemedicine technologies, which improves skill and confidence in engaging in Telemedicine [37].

Study participants who had more than years of work experience were about 3.1 times more ready for the telemedicine system as compared with those study participants who had less than five years of work experience. This is in line with a study done in Nigeria [38]. The possible reason might be that more working long years in the health care system might have the probability of sharing new knowledge with other partners in the workplace. The finding of this study could be generalizable to other private hospitals in Ethiopia.

This could be explained by the fact that those individuals, as new staff, might not have understood the context of the healthcare setting and the tiresome process flow imposed by paper-based records.

## The study's strengths and limitations

This is Ethiopia's first study to assess health professionals' readiness levels before Telemedicine implementation. It also highlighted some of the Measurements to be taken before the introduction of telemedicine in low-income countries. However, Because the study was cross-sectional, causality does not infer. The study did not triangulate with qualitative findings. Furthermore, this does not address organizational readiness.

## Conclusion and recommendation

Around two-thirds of the respondents had a good level overall of readiness for the adoption of telemedicine. The finding implied that less effort is required to improve readiness before the implementation of telemedicine. This finding implied that respondents who had good knowledge and a favorable attitude toward telemedicine were more ready for such technology. Capacity building is needed to enhance computer literacy, and computer skills build their confidence to rise ready for such technology. Building their capacity through training, building good internet connection, and availability of computers, where the necessary measures to improve Telemedicine readiness in this setting. Additionally, exploring healthcare providers' opinions with qualitative study and extending the proposed study to other implementation settings are recommended to be addressed in future works. The findings will help policymakers improve and expand the implementation of telemedicine technology across the country. The study has a positive impact on the successful implementation and use of telemedicine throughout hospitals at countries level by providing pertinent information about health professionals' preparedness status. Therefore, implementing telemedicine will have a significant contribution to the health system performance improvement in terms of providing quality care, accessibility to health facilities, reduction of costs, and creating a platform for communication between health professionals across different health institutions for providing quality patient care.

## Supporting information

**S1 Data.**
(SAV)

## Acknowledgments

We would like to acknowledge the hospital management for their support in carrying out this research. We would also want to acknowledge all the study participants and supervisors in the private hospitals who took part in this study.

## Author Contributions

**Conceptualization:** Sisay Maru Wubante.

**Data curation:** Sisay Maru Wubante.

**Formal analysis:** Sisay Maru Wubante, Araya Mesfin Nigatu.

**Funding acquisition:** Sisay Maru Wubante.

**Investigation:** Sisay Maru Wubante.

**Methodology:** Sisay Maru Wubante, Adamu Takele Jemere.

**Resources:** Sisay Maru Wubante.

**Software:** Sisay Maru Wubante.

**Supervision:** Sisay Maru Wubante.

**Validation:** Sisay Maru Wubante, Araya Mesfin Nigatu.

**Visualization:** Sisay Maru Wubante, Adamu Takele Jemere.

**Writing – original draft:** Sisay Maru Wubante.

**Writing – review & editing:** Sisay Maru Wubante, Araya Mesfin Nigatu.

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
