## [Decision Letter · Decision Letter 0]

14 Jul 2022

PONE-D-22-16282Health Professionals' readiness and its associated factors to implement Telemedicine system at private Hospitals in Amhara Region, Ethiopia 2021PLOS ONE

Dear Dr. Sisay wubante maru,

Thank you for submitting your manuscript to PLOS ONE. After careful consideration, we feel that it has merit but does not fully meet PLOS ONE’s publication criteria as it currently stands. Therefore, we invite you to submit a revised version of the manuscript that addresses the points raised during the review process.

Based on a thorough review of the manuscript and the evaluation of the reviewer's comments, our decision regarding the manuscript  is a major revision.

Major concerns:

The manuscript should be fundamentally revised from the typography and writing aspects.The introduction section needs to be rewritten and summarized. All abbreviations should be provided with their full equivalents the first time they are used.The discussion section needs to be summarized. Authors are expected to further discuss their results by comparing their results with similar studies. ==============================

We look forward to receiving your revised manuscript.

Kind regards,

Jahanpour Alipour, Ph.D.

Academic Editor

PLOS ONE

Journal Requirements:

2. In the ethics statement in the Methods, you have specified that verbal consent was obtained. Please provide additional details regarding how this consent was documented and witnessed, and state whether this was approved by the IRB.

 no. 

no competing interest. 

6. PLOS requires an ORCID iD for the corresponding author in Editorial Manager on papers submitted after December 6th, 2016. Please ensure that you have an ORCID iD and that it is validated in Editorial Manager. To do this, go to ‘Update my Information’ (in the upper left-hand corner of the main menu), and click on the Fetch/Validate link next to the ORCID field. This will take you to the ORCID site and allow you to create a new iD or authenticate a pre-existing iD in Editorial Manager. Please see the following video for instructions on linking an ORCID iD to your Editorial Manager account: https://www.youtube.com/watch?v=_xcclfuvtxQ.

7. Please ensure that you include a title page within your main document. You should list all authors and all affiliations as per our author instructions and clearly indicate the corresponding author.

8. Please amend either the abstract on the online submission form (via Edit Submission) or the abstract in the manuscript so that they are identical.

9. Your ethics statement should only appear in the Methods section of your manuscript. If your ethics statement is written in any section besides the Methods, please delete it from any other section. 

10. Please include a separate caption for each figure in your manuscript.

Reviewers' comments:

Reviewer's Responses to Questions

**Comments to the Author**

1. Is the manuscript technically sound, and do the data support the conclusions?

Reviewer #1: No

Reviewer #2: Yes

2. Has the statistical analysis been performed appropriately and rigorously? 

Reviewer #1: Yes

Reviewer #2: Yes

3. Have the authors made all data underlying the findings in their manuscript fully available?

Reviewer #1: No

Reviewer #2: Yes

4. Is the manuscript presented in an intelligible fashion and written in standard English?

Reviewer #1: No

Reviewer #2: Yes

5. Review Comments to the Author

Reviewer #1: Page 9: Paragraphs 1 and 2 which deal with the benefits of Telemedicine, should be more comprehensive and holistic

Page 9, lines 59-60: Some of these sentences are repetitive

Page 9, lines 65-66: A few words appear to have been missed.

Page 10: Cited studies from Lebanon, etc., lack sufficient details

Page 10: lines 86-90: This paragraph should be moved to the next page and merged with lines 108-113

Page 11. Lines 91-94: The sentence has no verb

Page 11, lines 108-109: Identify all five domains

Page 12, lines 112: what is the mean of dhis2? Is it a typo? Or is an abbreviation? If yes, it must be capitalized

The introduction is long and has repetitive sentences. The connection between the paragraphs and their logical order is not well established. The introduction needs a major rewrite. In addition, despite conducting the study during the Covid 19 pandemic, the Covid 19 pandemic and its impact on the spread of the use of telemedicine is not mentioned anywhere.

Line 136: Provide general information about ten hospitals, including number of beds, general or specialized, etc.

Page 13, line 141: Mention the number of specialists

Page 13, lines 151-162: In this part, too, information and repetitive sentences have been used a lot

Page 14, line 169: Specify which dimensions of health professional readiness are examined. In which part of the questionnaire is this dimension addressed and how many questions each part of the questionnaire have. The different parts of the questionnaire and the number of questions as well as the Likert scale used should be specified. Summarize the information presented in lines 197-211 and move to this section.

Page 15, line 176: what is the mean of" Three-degree holder health professionals"?

Line 178: It is better to explain in the introduction about the geographical extent of the study setting to show the importance of work.

Line 184: Reporting variables is not common in e-health studies.

Line 257: what is core and engagement readiness?

Line 244: Based on the data of which part of the questionnaire, this result was obtained?

Line 247: Are organizational factors just access to computers and the Internet? Numerous other factors had to be considered.

Figures 2 and 3 have no title and are not presented respectively

Line 264: Among the demographic variables, gender and educational status also had an effect on the level of readiness.

Lines 266-284: There is no need to duplicate table data in text format.

The Result section is not well organized because the method section does not specify the exact dimensions of the study and the questionnaire, and therefore the logical order of the findings is not clear to the reader.

Figures 2 and 3 have no title and are not presented respectively

Line 368: " Furthermore, another study also supports this study" how? Needs further explanation

The Discussion section needs to summarize comparisons to similar studies and place more emphasis on reasoning and inference.

There are several typos in reference section, for example references number 30 and 39

Reviewer #2: This work imed to assess health professionals' readiness and its determinant factors to adopte Telemedicine system at ten private hospitals in North West, Ethiopia. It would be interesting if they explained the rate between public and private hospitals in Ethiopia. Mainly for knowing the representativeness of the study among total professionals, both public and private.

They contend that those findings are not representative of the scenario in low-income countries due to differences in digital technology penetration and it needs several behavioral changes in the workplace for health workers and also knowing how to use computer technologies in day-to-day activities increase to use of telemedicine applications.

The results could be presented in a more summarized and less repetitive way.

6. PLOS authors have the option to publish the peer review history of their article (what does this mean?). If published, this will include your full peer review and any attached files.

Reviewer #1: No

Reviewer #2: **Yes: **Victoria Ramos

---

## [Author Response · Author response to Decision Letter 0]

8 Aug 2022

Dear Editors of PLOS ONE :

It has been recalled that we the authors of the manuscript entitled “Health Professionals' readiness and its associated factors to implement Telemedicine system at private Hospitals in Amhara Region, Ethiopia 2021” submitted our manuscript for publication in your journal and received reviewer comments for the betterment of the manuscript before its publication. In line with this, all authors are very happy with the constructive and valuable comments given by reviewers. Accordingly, we have considered all the comments and provided a point-by-point response and explanations for all the questions raised. Finally, we have submitted all the required documents in their revised form. We hope that we have addressed all the questions and if you have any points for further clarity, let us know.

All the authors would like to thank the editorial team and reviewers

Editor(s)’ comments to the authors 

Comment1: A rebuttal letter that responds to each point raised by the academic editor and reviewer(s). You should upload this letter as a separate file labeled 'Response to Reviewers'.

Answer: Thanks dear editor for your nice comments and suggestions. We, the authors

of this study, have attached the necessary files and a detailed rebuttal letter according to

your suggestion and the journal format.

Comment2: A marked-up copy of your manuscript that highlights changes made to the original version. You should upload this as a separate file labeled 'Revised Manuscript with Track Changes'.

Answer: Thank you, the track changes and cleaned document have been prepared and labeled as revised manuscript and attached

Comment 3: An unmarked version of your revised paper without tracked changes. You should upload this as a separate file labeled 'Manuscript'.

Answer: thank you very much, dear editor, unmarked version of the revised manuscript was prepared without track change labeled as the manuscript was uploaded.

Journal Requirements:

 Comments: When submitting your revision, we need you to address these additional requirements.

Answer: thank you dear editor we authors of this study agreed with your comments and suggestions. We have addressed the following requirements accordingly Plos one journal standards.

Comment 1: Please ensure that your manuscript meets PLOS ONE's style requirements, including those for file naming. The PLOS ONE style templates can be found at 

Answer: thank you for the insightful comments, we authors accepted all your comments and suggestions forwarded to our manuscript. Accordingly, the manuscript was prepared based on the Title, Author, Affiliations, and Manuscript body formatting guidelines of PLOS ONE journal.

Comment 2: In the ethics statement in the Methods, you have specified that verbal consent was obtained. Please provide additional details regarding how this consent was documented and witnessed, and state whether this was approved by the IRB.

Answer: thank you very much, we authors gladly accept your comments and suggestions. Before we collected the data, we inform the study participants about the objective of the study and study do not harm them when they participate and there are no incentives during participation in the study. After they give us their voluntarism to take part in the study we began collecting the data. This study was approved by an institutional review board of the University of Gondar college of medicine and health sciences institute of public health.

Comments 3. Thank you for stating the following financial disclosure: 

 no. 

Answer: thank you very much

Answer: thank you for the interesting suggestions, there are no specific funds for this study

b) State what role the funders took in the study. If the funders had no role in your study, please state: “The funders had no role in study design, data collection, and analysis, decision to publish, or preparation of the manuscript.”

Answer: thank you very much, no specific funding for this study

Answer: thank you no specific funding

Answer: thank you, we the authors of this study received no specific funding for this work.

Answer: thank you, our authors included amendments made in the cover letter

Comment 4: Thank you for stating the following in your Competing Interests section: 

no competing interest

Answer: thank you very much, the authors have declared that no competing interests exist."

Comment 5: In your Data Availability statement, you have not specified where the minimal data set underlying the results described in your manuscript can be found. PLOS defines a study's minimal data set as the underlying data used to reach the conclusions drawn in the manuscript and any additional data required to replicate the reported study findings in their entirety. All PLOS journals require that the minimal data set be made fully available. For more information about our data policy, please see http://journals.plos.org/plosone/s/data-availability.

Answer: thank you very much, dear editors, we authors of this study agreed on uploaded the dataset used in this study. We have uploaded additional supplementary materials.

Comment 6: PLOS requires an ORCID iD for the corresponding author in Editorial Manager on papers submitted after December 6th, 2016. Please ensure that you have an ORCID iD and that it is validated in Editorial Manager. To do this, go to ‘Update my Information’ (in the upper left-hand corner of the main menu), and click on the Fetch/Validate link next to the ORCID field. This will take you to the ORCID site and allow you to create a new iD or authenticate a pre-existing iD in Editorial Manager. Please see the following video for instructions on linking an ORCID iD to your Editorial Manager account: https://www.youtube.com/watch?v=_xcclfuvtxQ.

Answer: thank you very much, the corresponding author has ORCID ID and is validated by the editorial manager.

Comment7: Please ensure that you include a title page within your main document. You should list all authors and all affiliations as per our author instructions and indicate the corresponding author.

Answer: thank you, dear editors, for your interesting comments and suggestions and we authors of this study included title pages, a list of author’s affiliations, and corresponding authors accordingly to the PLoS one journal instructions.

 Comment 8: Please amend either the abstract on the online submission form (via Edit Submission) or the abstract in the manuscript so that they are identical.

Answer: thank you for your insightful comments and suggestions, we agreed on the identity of the abstracts and .accordingly revised the abstracts online.

 Comment 9: Your ethics statement should only appear in the Methods section of your manuscript. If your ethics statement is written in any section besides the Methods, please delete it from any other section.

Answer: thank you very much, we authors of this study are grateful for the comments and suggestions forward. we revised accordingly and the ethics statement was put in method sections.

 Comment 10: Please include a separate caption for each figure in your manuscript.

Answer: thank you, we have given separate captions for each figure that appears in the revised manuscript.

Review1 Comments to the Author

Reviewer #1: Page 9: Paragraphs 1 and 2 which deal with the benefits of Telemedicine, should be more comprehensive and holistic.

Answer: thank you very much for the interesting suggestions and comments, we authors agreed with your fantastic comments. Accordingly revised in the main manuscript is shown in track change.

Reviewer #1:Page 9, lines 59-60: Some of these sentences are repetitive

Answer: thank you for the interesting and constructive comments and suggestions; we authors of this study gladly accepted all comments and suggestions. Thus all repetitive sentences were removed in manuscript track changes.

Reviewer #1: Page 9, lines 65-66: A few words appear to have been missed.

Answer: thank you very much, we authors of the study accepted all your comments and suggestions. In addition, we have added missed words in the revised track change manuscript.

Reviewer #1: Page 10: Cited studies from Lebanon, etc., lack sufficient details

Answer: thank you, dear reviewer, we agreed on your advisable comments and suggestion forwarded to us .accordingly we try to write more clearly and in detail in the track change manuscript. 

Reviewer #1:Page 10: lines 86-90: This paragraph should be moved to the next page and merged with lines 108-113

Answer: thank you very much, we authors of this study accepted all your comments and suggestions. According to your suggestions, we have moved the paragraphs and put them in the appropriate place in the track change manuscript.

Reviewer #1: Page 11. Lines 91-94: The sentence has no verb

Answer: thank you for the insightful suggestions and comments forwarded to our manuscript. Our authors accepted all comments and revised accordingly in the track change manuscript.

Reviewer #1:Page 11, lines 108-109: Identify all five domains

Answer: thank you very much, we accepted all comments and suggestions. Accordingly, the five domains were added in the rack change revised manuscript.

Reviewer #1: Page 12, lines 112: what is the mean of dhis2? Is it a typo? Or is it an abbreviation? If yes, it must be capitalized

Answer: thank you very much, we authors accepted comments and suggestions. Accordingly revised in the track changes manuscript. it was an abbreviation for district health information system version 2.

Reviewer #1: The introduction is long and has repetitive sentences. The connection between the paragraphs and their logical order is not well established. The introduction needs a major rewrite. In addition, despite conducting the study during the Covid 19 pandemic, the Covid 19 pandemic and its impact on the spread of the use of telemedicine is not mentioned anywhere.

Answer: thank you very much for your insightful comments and suggestions, we gladly accepted suggestions and comments. According to suggestions we try to reduce repetitive and long sentences and establish logical order between paragraphs. in the era of covid-19 pandemics, we try to show the importance of telemedicine. 

Reviewer #1: Line 136: Provide general information about ten hospitals, including the number of beds, general or specialized, etc.

Answer: thank you very much, we authors of this study accepted all suggestions and comments. Accordingly, we incorporate basic information about private hospitals in the revised track change manuscript.

Reviewer #1: Page 13, line 141: Mention the number of specialists

Answer: thank you dear reviewers for the insightful constructive comments and suggestions; we gladly accepted all comments. We try to incorporate the numbers of specialists working in each hospital in the revised track change manuscript.

Reviewer #1: Page 13, lines 151-162: In this part, too, information and repetitive sentences have been used a lot.

Answer: thank your comments and suggestions forwarded to our manuscript development are very interesting and advisable for us. We tried to reduce too much information and redundancy appeared in this part in the revised tracked change manuscript.

Reviewer #1: Page 14, line 169: Specify which dimensions of health professional readiness are examined. In which part of the questionnaire is this dimension addressed and how many questions does each part of the questionnaire have. The different parts of the questionnaire and the number of questions, as well as the Likert scale used, should be specified. Summarize the information presented in lines 197-211 and move to this section.

Answer: thank you, we accepted all comments and suggestions. in this study, we try to assess core and engagement readiness of telemedicine among health professionals .likert type questions were used to assess readiness. A total of 15 questions four questions for core readiness and 11 questions for engagement were used. This information is summarized and presented accordingly to the suggestions in the revised track change manuscript.

Reviewer #1: Page 15, line 176: what is the meaning of" Three-degree holder health professionals"?

Answer: thank you for the comments and suggestions we authors of this study agreed on the issues raised. We mean that data collection supervisors were bachelor's degree health informatics professionals. Try to revise in track change revised manuscript.

Reviewer #1: Line 178: It is better to explain in the introduction about the geographical extent of the study setting to show the importance of work.

Answer: thank you for your insightful comments and suggestions, we authors are happy to accept comments and based on your recommendation try to remove them from this part in the track change manuscript.

Reviewer #1: Line 184: Reporting variables is not common in e-health studies

Answer: thank you for the comments forwarded to our manuscript improvement. We put the study variables assuming for clarification purposes. However if not necessary we tried to remove Based on your recommendation on track change manuscript.

Reviewer #1: Line 257: what is core and engagement readiness?

Answer: thank you reviewer the comments given us to improve the scientific writing of our manuscript. We try to give clear information about core and engagement readiness in the methods section in the track change manuscript. 

Reviewer #1:Line 244: Based on the data which part of the questionnaire, this result was obtained?

Answer: thank you very much interesting comments on the advancement of our manuscript quality and readability. Accordingly, the result was obtained by technical-related factors questions. we tried to add a table for this result look (table4).

Reviewer #1: Line 247: Are organizational factors just access to computers and the Internet? Numerous other factors had to be considered.

Answer: thank you for the constructive comments and suggestions forwarded to us to advance the manuscript writing and understandable. We accepted all your comments, internet, and computer access are not the only organizational factors that have positive or negative impacts on health professionals' readiness we try to incorporate the availability of technical personnel and power generator but internet and computer access were significantly associated with health professionals readiness.

Reviewer #1: Figures 2 and 3 have no title and are not presented respectively

Answer: thank you very much, we authors of this study accepted all your comments and suggestions to improve our manuscript scientific writing. Accordingly, we tried to give a full name for figure 2 and figure 3 in the revised track change manuscript.

Reviewer #1: Line 264: Among the demographic variables, gender and educational status also affected the level of readiness.

Answer: thank you, reviewer, for constructive comments, we accepted all raised suggestions and comments to clarify our manuscript readability and scientific writing .accordingly we tried to incorporate the missed socio-demographic variables including gender and educational status of health professionals in the revised track change manuscript.

Reviewer #1: Lines 266-284: There is no need to duplicate table data in text format.

Answer: thank you very much for the interesting comments forwarded to our manuscript improvement. We accepted all your comments. The table data was converted into meaningful information for readers’ understandable concept .but your recommendation is that duplication, so we tried to reduce the text information written in an articulated manner in the revised track change manuscript.

Reviewer #1: The Result section is not well organized because the method section does not specify the exact dimensions of the study and the questionnaire, and therefore the logical order of the findings is not clear to the reader

Answer: thank you very much reviewer, we accepted all your comments and suggestions given to us to advance scientific writing in the manuscript. We tried to revise the method sections information to make the logical order of the manuscript clear for readers in track change manuscript

Reviewer #1: Figures 2 and 3 have no title and are not presented respectively

Answer: thank you very much, we authors of this study accepted all your comments and suggestions to improve our manuscript scientific writing. Accordingly, we tried to give full names for Figures 2 and figure 3 in the revised track change manuscript.

Reviewer #1: Line 368: “Furthermore, another study also supports this study" how? Needs further explanation

Answer: thank you very much for your insightful comments on our manuscript development, we authors accepted all comments forwarded to us. Accordingly, we tried to revise and modify the section in the track change document. 

Reviewer #1: The Discussion section needs to summarize comparisons to similar studies and place more emphasis on reasoning and inference.

Answer: thank you, reviewer, we accepted all your comments and suggestions forwarded to our manuscript improvement. Accordingly, we revised the discussion section thoroughly in the track change manuscript.

Reviewer #1: There are several typos in the reference section, for example, references number 30 and 39

Answer: thank you for your insightful comments on our manuscript development, we accepted all comments. we tried to correct typos in the reference section in the revised track change manuscript.

Reviewer2 Comments to the Author

Reviewer #2: This work aimed to assess health professionals' readiness and the determinant factors to adopt the Telemedicine system at ten private hospitals in North West, Ethiopia. It would be interesting if they explained the rate between public and private hospitals in Ethiopia. Mainly for knowing the representativeness of the study among total professionals, both public and private.

Answer: thank you very much for your insightful comments on our manuscript improvement readability and scientific writing. We authors of this study accepted all comments, unfortunately, this study examined the proportion of health professionals who are working at private hospitals. Due to this public hospitals' health professional’s proportion of readiness is not examined. 

Reviewer #2: They contend that those findings are not representative of the scenario in low-income countries due to differences in digital technology penetration and it needs several behavioral changes in the workplace for health workers and also know how to use computer technologies in day-to-day activities to increase to the use of telemedicine applications.

Answer: we thank you reviewer for your interesting comments and suggestions, we accepted all comments and suggestions given to our manuscript improvement.

Reviewer #2: The results could be presented in a more summarized and less repetitive way.

Answer: thank you very much for your insightful comments on this paper improvement, we accepted all comments and suggestions. Accordingly revised in the track change manuscript.

---

## [Decision Letter · Decision Letter 1]

7 Sep 2022

PONE-D-22-16282R1Health Professionals' readiness and its associated factors to implement Telemedicine system at private Hospitals in Amhara Region, Ethiopia 2021PLOS ONE

Dear Sisay wubante maru,

Thank you for submitting your manuscript to PLOS ONE. After careful consideration, we feel that it has merit but does not fully meet PLOS ONE’s publication criteria as it currently stands. Therefore, we invite you to submit a revised version of the manuscript that addresses the points raised during the review process.

We look forward to receiving your revised manuscript.

Kind regards,

Jahanpour Alipour, Ph.D.

Academic Editor

PLOS ONE

Journal Requirements:

Reviewers' comments:

Reviewer's Responses to Questions

**Comments to the Author**

1. If the authors have adequately addressed your comments raised in a previous round of review and you feel that this manuscript is now acceptable for publication, you may indicate that here to bypass the “Comments to the Author” section, enter your conflict of interest statement in the “Confidential to Editor” section, and submit your "Accept" recommendation.

Reviewer #1: (No Response)

Reviewer #2: All comments have been addressed

2. Is the manuscript technically sound, and do the data support the conclusions?

Reviewer #1: Yes

Reviewer #2: Yes

3. Has the statistical analysis been performed appropriately and rigorously? 

Reviewer #1: Yes

Reviewer #2: Yes

4. Have the authors made all data underlying the findings in their manuscript fully available?

Reviewer #1: Yes

Reviewer #2: Yes

5. Is the manuscript presented in an intelligible fashion and written in standard English?

Reviewer #1: No

Reviewer #2: Yes

6. Review Comments to the Author

Reviewer #1: In conclusion, the macro view should be observed and the impact of this research on the successful use of telemedicine and improvement of the health system should be mentioned.

In order to improve the validity of the sentence "In the era of covid-19 pandemics

telemedicine have a great role in controlling the pandemics by using different communication …." in the introduction section (lines 66-69) on page 15, refer to the following reference.

https://pubmed.ncbi.nlm.nih.gov/34528234/

Opportunities and Challenges of Telehealth in Disease Management during COVID-19 Pandemic: A Scoping Review

Reviewer #2: The authors have adequately addressed my comments raised in a previous round of review and I believe that this manuscript is now acceptable for publication,

7. PLOS authors have the option to publish the peer review history of their article (what does this mean?). If published, this will include your full peer review and any attached files.

Reviewer #1: **Yes: **Mohammad Hosein Hayavi-Haghighi

Reviewer #2: No

---

## [Author Response · Author response to Decision Letter 1]

8 Sep 2022

Dear Editors of PLOS ONE :

It has been recalled that we the authors of the manuscript entitled “Health Professionals' readiness and its associated factors to implement Telemedicine system at private Hospitals in Amhara Region, Ethiopia 2021” submitted our manuscript for publication in your journal and received reviewer comments for the betterment of the manuscript before its publication. In line with this, all authors are very happy with the constructive and valuable comments given by reviewers. Accordingly, we have considered all the comments and provided a point-by-point response and explanations for all the questions raised. Finally, we have submitted all the required documents in their revised form. We hope that we have addressed all the questions and if you have any points for further clarity, let us know.

All the authors would like to thank the editorial team and reviewers

Editor(s)’ comments to the authors 

Comment1: A rebuttal letter that responds to each point raised by the academic editor and reviewer(s). You should upload this letter as a separate file labeled 'Response to Reviewers'.

Answer: Thanks dear editor for your nice comments and suggestions. We, the authors

of this study, have attached the necessary files and a detailed rebuttal letter according to

your suggestion and the journal format.

Comment2: A marked-up copy of your manuscript that highlights changes made to the original version. You should upload this as a separate file labeled 'Revised Manuscript with Track Changes.

Answer: Thank you, the track changes and cleaned document have been prepared and labeled as revised manuscript and attached

Comment 3: An unmarked version of your revised paper without tracked changes. You should upload this as a separate file labeled 'Manuscript'.

Answer: thank you very much, dear editor, unmarked version of the revised manuscript was prepared without track change labeled as the manuscript was uploaded.

Journal Requirements:

 Comments: Please review your reference list to ensure that it is complete and correct. If you have cited papers that have been retracted, please include the rationale for doing so in the manuscript text, or remove these references and replace them with relevant current references. Any changes to the reference list should be mentioned in the rebuttal letter that accompanies your revised manuscript. If you need to cite a retracted article, indicate the article’s retracted status in the References list and also include a citation and full reference for the retraction notice.

Answer: thank you dear editor we authors of this study agreed with your comments and suggestions. We tried to review the whole references cited in the manuscript are complete and correct. we ensured that all references are correct and complete no retracted papers are cited in our manuscript.

Reviewer 1 comment to the author

Comment#reviwer1: In conclusion, the macro view should be observed and the impact of this research on the successful use of telemedicine and the improvement of the health system should be mentioned.

Answer: thank you very much for your valuable comments ad suggestion for the improvement of our manuscript, we authors are happy to accept and agreed with the comments. Accordingly, we revised the conclusion section as recommended in the main revised manuscript.

Comment#reviwer1: To improve the validity of the sentence "In the era of covid-19 pandemics telemedicine have a great role in controlling the pandemics by using different communication …." in the introduction section (lines 66-69) on page 15, refer to the following reference.

Answer: thank you very much, dear reviewer, for your insightful comments and suggestion for our manuscript development. We authors of this study gladly accepted and agreed with your comments. We tried to cite the recommended reference in the introduction section lines66-69 in the revised manuscript.

Reviwer2 comments to the Author

Reviewer #2: The authors have adequately addressed my comments raised in a previous round of review and I believe that this manuscript is now acceptable for publication,

Answer: thank you very much dear reviewer for your insightful appreciation and recognition, we authors of this study gladly accepted and admiration for your constructive scientific comments forwarded to our manuscript improvement. Thank you very much again.

---

## [Editor Report · Decision Letter 2]

12 Sep 2022

Health Professionals' readiness and its associated factors to implement Telemedicine system at private Hospitals in Amhara Region, Ethiopia 2021

PONE-D-22-16282R2

Dear Dr. Sisay wubante maru,

We’re pleased to inform you that your manuscript has been judged scientifically suitable for publication and will be formally accepted for publication once it meets all outstanding technical requirements.

Kind regards,

Jahanpour Alipour, Ph.D.

Academic Editor

PLOS ONE

Additional Editor Comments (optional):

The authors properly addressed satisfactorily all of the concerns. Thus, the manuscript has now could be accepted for publication in the PLOS ONE journal.
---

## [Editor Report · Acceptance letter]

15 Sep 2022

PONE-D-22-16282R2 

Health Professionals' readiness and its associated factors to implement Telemedicine system at private Hospitals in Amhara Region, Ethiopia 2021 

Dear Dr. Wubante:

I'm pleased to inform you that your manuscript has been deemed suitable for publication in PLOS ONE. Congratulations! Your manuscript is now with our production department. 

Kind regards, 

on behalf of

Dr., Jahanpour Alipour 

Academic Editor

PLOS ONE